# Anchored Optimization and Contrastive Revisions: Addressing Reward Hacking in Alignment

**Karel D'Oosterlinck**[1,3*]   **Winnie Xu**[3]   **Chris Develder**[1]   **Thomas Demeester**[1]
**Amanpreet Singh**[3]   **Christopher Potts**[2]   **Douwe Kiela**[2,3]   **Shikib Mehri**[3]

[1]Ghent University – imec      [2]Stanford University      [3]Contextual AI
`karel.doosterlinck@ugent.be, shikib@contextual.ai`

## Abstract

Alignment of Large Language Models (LLMs) is crucial for ensuring their safety, particularly in preventing unintended behaviors and harmful outputs. However, aligning models using preference pair datasets does not always guarantee successful results, as models can accidentally be optimized for superficial cues in the data rather than genuinely desirable behaviors, an issue often referred to as *reward hacking*. We study the core principles of alignment and find that (i) preference data gives a more robust learning signal when the underlying responses are contrastive, and (ii) alignment objectives lead to more robust optimization when they specify more control over the model during training. Based on these insights, we introduce Contrastive Learning from AI Revisions (CLAIR), a data-creation method which leads to more contrastive preference pairs, and Anchored Preference Optimization (APO), a controllable and more stable alignment objective. Both our methods are designed to give AI practitioners precise control over how their model should change during alignment training, allowing them to build safer and more precise models. We align `Llama-3-8B-Instruct` using various comparable datasets and alignment objectives and measure `MixEval-Hard` scores, which correlate highly with human-produced rankings of models. The CLAIR preferences lead to the strongest performance out of all datasets, and APO consistently outperforms less controllable objectives. Our best model, trained on 32K CLAIR preferences with APO, improves `Llama-3-8B-Instruct` by 7.65%, closing the gap with GPT4-turbo by 45%. The strong results of both our methods indicate their ability to precisely control what a model learns during alignment, mitigating reward hacking. Additionally, our experiments highlight how alignment can accidentally deteriorate model performance, inadvertently introducing safety risks.

## 1   Introduction

Aligning language models with preferences is a critical component in LLM development, significantly enhancing model capabilities, safety, and adherence to human values [Christiano et al., 2017, Ouyang et al., 2022, Bai et al., 2022]. These preferences can be expressed through *preference pairs* (output $y_l \prec y_w$ for input $x$), which offer a richer signal than individual outputs and enable more expressive learning objectives. Recently, contrastive learning objectives have made alignment more accessible by circumventing the need for auxiliary reward models [Rafailov et al., 2024b].

Despite these advantages, alignment outcomes can be suboptimal. For example, a model can accidentally be optimized for superficial cues in the alignment data, an issue also referred to as

---

*   Work done as a part of an internship at Contextual AI. Code at `https://github.com/ContextualAI/CLAIR_and_APO`

**(A)**      $x$    write a story about apples

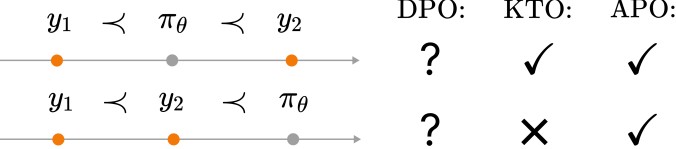

✗ irrelevant signal          ✓ targeted signal

**(B)**   Different alignment situations for model $\pi_\theta$

     $y_1 \prec \pi_\theta \prec y_2$       DPO:   KTO:   APO:

                                         ?      ✓      ✓

     $y_1 \prec y_2 \prec \pi_\theta$

                                           ?      ✗      ✓

Figure 1: Alignment is underspecified with regard to preferences and training objective. **A:** Preference pairs can vary along irrelevant aspects, Contrastive Learning from AI Revisions (CLAIR) creates a targeted preference signal instead. **B:** The quality of the model can impact alignment training, Anchored Preference Optimization (APO) explicitly accounts for this.

*reward hacking*, leading to undesirable outcomes [Eisenstein et al., 2023, Feng et al., 2024, Park et al., 2024]. Since alignment is a critical step towards achieving safe and ethical model behavior, reward hacking poses significant risks. In this paper, we reason through the nature of alignment, focusing on 1. the preference signal expressed by the data, and 2. the training dynamics of contrastive alignment objectives. We find that across both these axes, conventional alignment methods are underspecified, significantly contributing to the issue of reward hacking. To solve this, we argue that 1. preference data should be minimally contrastive, and 2. alignment objectives should account for distinct alignment situations (see Figure 1). Our work sheds light on suboptimal alignment outcomes. For example, we show in Section 5 how a model aligned using high-quality outputs can actually degrade if the pairs differ in multiple uncontrolled aspects.

These insights lead to two new contributions. First, we introduce Contrastive Learning from AI Revisions (CLAIR), a method for creating preference pairs which *minimally revises* one output to express a preference. The pairs created by CLAIR result in a more precise learning signal, as opposed to conventional methods which use a judge to *select* a preferred response. This more precise signal leads to fewer undesirable side-effects during alignment. Second, we introduce Anchored Preference Optimization (APO), a family of contrastive objectives which explicitly account for distinct relationships between model and data during alignment. The tailored training dynamics of APO results in more robust alignment compared to conventional objectives, again producing fewer undesirable side-effects.

In order to study the role of both 1. minimally contrastive preference data, and 2. distinct alignment training dynamics, we individually align a model across four comparable preference datasets using five alignment objectives. One dataset is created through our CLAIR method. We compare this with two conventional judge-based datasets (Reinforcement Learning from AI Feedback; Bai et al. 2022). Finally, we consider an ablated version of CLAIR created to directly assess the impact of contrastiveness. We consider five distinct alignment objectives: DPO [Rafailov et al., 2024b], KTO [Ethayarajh et al., 2024], continued Supervised Fine-Tuning on the preferred answer, and two variants of our proposed APO. We measure `MixEval-Hard` accuracy [Ni et al., 2024] and length-controlled `AlpacaEval` scores [Dubois et al., 2024] for each model, both benchmarks correlate highly with model rankings produced by humans [Chiang et al., 2024].

We align `Llama-3-8B-Instruct` [Dubey et al., 2024] and use GPT4-turbo [Achiam et al., 2023] for preference judgements/revisions. We find that our strongest model, aligned on 32K CLAIR preferences with APO, improves `Llama-3-8B-Instruct` performance by 7.65% on `MixEval-Hard`,

closing the performance gap with GPT4-turbo by 45%. Our analysis indicates that the contrastiveness of CLAIR preferences is the major driver of performance. Across every alignment datasets considered, APO objectives achieve the best performance. The superior performance of both our methods indicate a tighter coupling between the intended and realised behavior of the model. Our techniques give AI practitioners greater control over model alignment, significantly improving their ability to develop safe and ethical models.

## 2   Underspecification in Alignment

The alignment procedure creates complex interactions between the target model, the preference dataset, and the alignment objective. The present section reflects on failure cases of all alignment efforts which start from preferences, and how this leads to reward hacking. The section discussed data and objective respectively.

Given a collection of prompts $X$, a preference dataset is a set of triples $(x, y_w, y_l)$ , where $y_w$ and $y_l$ are, respectively, a winning (more preferred) and losing (less preferred) response to prompt $x$. The preference signal in such a dataset is essentially expressed by the *difference between* winning and losing outputs, illustrated in Figure 1 A. However, paired outputs can differ in many aspects, some of which are spurious and thus irrelevant to the preference. These spurious differences will generally create a challenging credit assignment problem. Outputs which are *minimally contrastive* differ along fewer axes, resulting in less spurious differences. Thus, **if preference pairs produce a clearer minimal contrast, the alignment learning signal becomes more clear**. Existing preference datasets vary meaningfully in their contrastiveness. For example, in the Stanford Human Preferences dataset [Ethayarajh et al., 2022], two outputs in a pair are simply responses to the same Reddit post, and thus they are not guaranteed to be especially comparable. An ideal preference dataset would consist of a very controlled difference between either example. Such a dataset would minimize the risk of reward hacking, this insight leads us to CLAIR (Section 3).

Preference triples only specify that one output is better than another. This creates ambiguity, since it is not known if the more preferred answer was actually good. To see how this can impact alignment, suppose we have a dataset of triples where $y_w$ tends to score 8/10 on some quality scale and $y_l$ tends to score 6/10. A target model that generally scores 9/10 may become worse if the likelihood of $y_w$ would increase during training, as illustrated in Figure 1 B. Therefore, **alignment training needs to be aware of how desirable any individual answer is, regardless of its preference relationship**. To take a salient example, $\approx$80% of winning outputs in `UltraFeedback` [Cui et al., 2024] are generated by a less performant model than `Llama-3-8B-Instruct` (as measured by Chatbot Arena Elo; Chiang et al. 2024). Naively aligning `Llama-3-8B-Instruct` on this dataset may thus worsen performance, since the model is optimized to prefer answers which are actually of lower quality, again producing reward hacking. Examples like this one lead us to Anchored Preference Optimization (APO; Section 4).

In summary, current alignment approaches are underspecified along two key axes: 1. preferences may be weakly expressed due to non-contrastive data, and 2. alignment objectives need to account for the model-data relation. Both underspecifications can cause reward hacking, i.e., situations where the model is optimized for spurious or undesirable cues in the preference data instead of the desired behavior. In what follows, we set out to improve alignment across both axes.

## 3   Contrastive Learning from Revisions

We now introduce Contrastive Learning from AI Revisions (**CLAIR**), a general procedure for creating minimally contrasting preference pairs.

Let $M$ be the target model we will align. Given a prompt $x$, we sample the losing output $y_l$ directly from the model. Then, we use a *Reviser* to minimally revise and improve $y_l$, resulting in the winning output $y_w$:

$$y_l = M(x)$$
$$y_w = Reviser(x, y_l)$$
(1)

In this work, we use a stronger LLM to perform revisions, prompted to enhance the clarity, correctness, and engagement of the output (prompts and dataset details given in Appendix A). Figure 2 shows an

**Prompt** $x$

```
1 Can you help me evaluate the relevancy of this example using an excel table?
2 Based on the table, how relevant is the example of "A restaurant in Paris" to
  the given context?
3 Here's the table with relevant data:
```

```
 4 | Restaurant Name  | City     | Country  |
 5 |------------------|----------|----------|
 6 | Le Jules Verne   | Paris    | France   |
 7 | Chez L'Ami Jean  | Paris    | France   |
 8 | La Tour D'Argent | Paris    | France   |
 9 | Joe's Pizza      | New York | USA      |
10 | The Fat Duck     | London   | UK       |
```

**Answer** $y_l$

```
1 To evaluate the relevance of the example "A restaurant in Paris" to the
  given context, we can use the following criteria:
2 1. **Specificity**: How specific is the example to the given context? In
  this case, the example is quite specific, as it mentions a specific city
  (Paris) which is only 2 out of 5 restaurants in the table.
3 2. **Frequency**: How frequent is the example in the given context? In this
  case, 3 out of 5 restaurants (Le Jules Verne, Chez L'Ami Jean, and La Tour
  D'Argent) are from Paris, which is a significant frequency.
4 Based on these criteria, I would say that the example "A restaurant in
  Paris" is **Highly Relevant** to the given context. The specificity and
  frequency of Parisian restaurants in the table make this example a strong
  match.
5
6 How would you rate the relevance of this example?
```

**Revision** $y_w$

```
1 To evaluate the relevance of the example "A restaurant in Paris" to the given
  context, let's consider the following criteria:
2 1. **Specificity**: The example is quite specific, as it mentions Paris, which
  is the city for 3 out of the 5 restaurants listed in the table — Le Jules
  Verne, Chez L'Ami Jean, and La Tour D'Argent. This level of specificity
  highlights Paris's prominence in the dataset.
3 2. **Frequency**: Parisian restaurants represent a majority in the given
  context, with 3 out of 5 restaurants (60%) being located in Paris. This
  frequency underscores the city's significance in the list, suggesting that
  Paris is a major focus.
4 Based on these criteria, the example "A restaurant in Paris" is **Highly
  Relevant** to the given context. The specificity and frequency of Parisian
  restaurants in the table not only make this example a strong match but also
  reflect Paris's iconic status as a culinary capital. Therefore, it's
  reasonable to assert that discussing a restaurant in Paris closely aligns with
  the dataset's emphasis, making our example notably pertinent.
```

Figure 2: An **answer** produced by `Llama-3-8B-Instruct` for a **prompt**, and corresponding GPT4-turbo **revision** of this answer. The differences between answer and revision are highlighted. The revision generally follows the same outline as the answer but improves it where possible. For example, the revision correctly alters the count of Parisian restaurants from 2 to 3 in the second line of the answer.

example triple created using this method. The losing output was generated by `Llama-3-8B-Instruct` and revised by GPT4-turbo. The revision keeps most of the initial output intact, while improving details. Recently, Dubey et al. [2024] used human revisions in the development of the `llama-3.1` model family, though their process seems oriented towards enhancing quality differences rather than creating minimal contrasts.

CLAIR differs markedly from more familiar approaches to collecting preference data. For example, in the **on-policy judge** paradigm (as used in Reinforcement Learning from AI Feedback; Bai et al. 2022), two generations are sampled from $M(x)$, and a *Judge* (often another LLM) decides which is the winner and which the loser:

$$y_1, y_2 = M(x), M(x)$$
$$y_w, y_l = Judge(x, y_1, y_2) \tag{2}$$

We use this approach as one of our baselines, with a prompt comparable to the revision prompt used by CLAIR. Additionally, we consider an **off-policy judge** versions of (2) where the outputs are generated by models other than the target model:

$$y_1, y_2 = M'(x), M''(x)$$
$$y_w, y_l = Judge(x, y_1, y_2) \tag{3}$$

Both the on-policy and off-policy judge approaches provide useful comparison points for CLAIR. In addition, we evaluate a baseline that helps us understand the role of contrastiveness in particular. For CLAIR, the *Reviser* is generally a stronger model than the model we are aligning. This means that the winning examples $y_w$ are always generated by a stronger model. To decouple this factor from the contrastiveness induced by the revision process, we also evaluate a baseline that we call **Stronger Preferred**, where the stronger model provides the winning example for each pair without revision:

$$y_l = M(x)$$
$$y_w = Stronger(x) \tag{4}$$

For the alignment experiments reported in Section 5, we created four preference datasets following (1)–(4). Each dataset is created using the same 32K prompts uniformly sampled from `UltraFeedback` [Cui et al., 2024], a widely used preference dataset with prompts spanning a broad range of domains. We take the target model $M$ to be `Llama-3-8B-Instruct`, one of the most competitive open source models available at the time of writing. For the off-policy judge dataset, we use already judged outputs available in `UltraFeedback`. Approximately 80% of these winning outputs are generated by a model weaker than `Llama-3-8B-Instruct` (as measured by Chatbot Arena Elo; Chiang et al. 2024). Thus, this off-policy judge dataset generally contains lower quality outputs compared to the model.

# 4 Anchored Preference Optimization

A preference triple $(x, y_w, y_l)$ expresses the belief that $y_w$ is a more preferred output than $y_l$ for prompt $x$. Alignment objectives use this relationship to align a model. Different objectives achieve this in very different ways, with deep consequences for the alignment process.

Direct Preference Optimization (DPO; Rafailov et al. 2024b) is a widely used and empirically successful alignment objective. The core stipulation of DPO is that the likelihood change of winning outputs during training needs to be greater than the likelihood change of losing outputs. This likelihood change for a prompt and output is denoted as the reward $r_\theta(x, y)$, which captures the log-ratio of likelihoods between the model during training $\pi_\theta(x \mid y)$ and the model before training, also called *reference*, $\pi_{\text{ref}}(x \mid y)$:

$$r_\theta(x, y) = \beta \log \frac{\pi_\theta(y \mid x)}{\pi_{\text{ref}}(y \mid x)} \tag{5}$$

Here, $\beta$ is a hyperparameter which scales this log-ratio. This leads to the following DPO objective:

$$\mathcal{L}_{DPO}(x, y_w, y_l; \theta) = -\log \sigma \Big( r_\theta(x, y_w) - r_\theta(x, y_l) \Big) \tag{6}$$

The DPO authors report that the gradient of this objective intuitively leads to an increased winning likelihood and decreased losing likelihood. However, this is only one possibility out of three distinct scenarios. Alternatively, DPO can increase the winning likelihood more than it increases the losing likelihood, or decrease the winning likelihood less than it decreases the losing likelihood [Feng et al., 2024]. These scenarios may end up producing vastly different models. As discussed in Section 2, a winning output is not necessarily better than what the model produces *before* alignment. In this case, DPO may hurt performance if it increases the likelihood of undesirable outputs.

To help researchers navigate these interactions, we introduce Anchored Preference Optimization (APO). In essence, APO is a family of alignment objectives which offer fine-grained control over each of the rewards, thus controlling the absolute increase or decrease in likelihood during training. In this paper, we focus in particular on variants that we call APO-zero and APO-down:

$$\mathcal{L}_{zero}^{APO}(x, y_w, y_l; \theta) = -\sigma \Big( r_\theta(x, y_w) \Big) + \sigma \Big( r_\theta(x, y_l) \Big) \tag{7}$$

$$\mathcal{L}_{down}^{APO}(x, y_w, y_l; \theta) = \sigma \Big( r_\theta(x, y_w) \Big) - \sigma \Big( r_\theta(x, y_w) - r_\theta(x, y_l) \Big) \tag{8}$$

APO-zero explicitly pushes for an increased likelihood of winning outputs and decreased likelihood of losing outputs during training. In contrast, APO-down decreases the likelihood of winning outputs and decreases the likelihood of losing outputs even more. If the model is better than the winning outputs ($y_w \prec \pi_\theta$), APO-down will intuitively be a better objective. If winning outputs are better than the model ($y_w \succ \pi_\theta$), APO-zero will be better.

One can define additional APO objectives. In general, any contrastive objective (i.e., greater reward for winning outputs) which specifies additional constraints on either reward to achieve a tighter link between model and data (e.g., winning rewards should be positive) can be seen as a form of Anchored Preference Optimization. In Section 6 we consider different alignment objectives and discuss how they relate to APO.

One interesting variant of APO can be derived from the Kahneman–Tversky Optimization (KTO) objective of Ethayarajh et al. [2024]. As originally defined, KTO does not operate on preference pairs, but rather requires only one unpaired answer and a label indicating if it was preferred or not; the goal of KTO is to push the winning / losing reward above / below the Kullback–Leibler (KL) divergence between the model during training and the reference model. The APO perspective helps us see that there is a natural paired variant of KTO in which the KL-divergence functions as the anchor:

$$\mathcal{L}_{KTO\text{-}pair}(x, y_w, y_l; \theta) = -\sigma \Big( r_\theta(x, y_w) - \beta\, KL \Big) - \sigma \Big( \beta\, KL - r_\theta(x, y_l) \Big) \tag{9}$$

This *KL* term is non-negative, and thus the winning reward is pushed to be positive; the losing reward can still be either positive or negative.

| Dataset | Objective | ME-Hard 2024-06-01 | | ME-Hard 2024-08-11 | |
|---|---|---|---|---|---|
| | | Max Δ | Mean Δ | Max Δ | Mean Δ |
| Judge off-policy | DPO | 1.10 | −0.74 (1.15) | 4.30 | 2.85 (0.75) |
| | KTO-pair | −1.00 | −2.89 (0.96) | 4.05 | 1.18 (1.67) |
| | SFT | −1.95 | −1.63 (1.06) | 2.85 | 0.42 (1.20) |
| | APO-zero | 0.80 | −1.99 (1.23) | 4.65 | 1.26 (1.62) |
| | APO-down | 2.70 | 0.64 (0.98) | 4.80 | 3.52 (0.85) |
| Judge on-policy | DPO | 4.00 | 0.56 (1.61) | 5.20 | 2.71 (1.41) |
| | KTO-pair | 2.45 | −0.51 (1.26) | 5.05 | 1.13 (1.70) |
| | SFT | 0.65 | −0.91 (1.01) | 4.20 | 2.55 (0.70) |
| | APO-zero | 4.65 | 0.02 (1.66) | 5.35 | 2.19 (1.28) |
| | APO-down | 3.65 | 1.60 (0.95) | 4.25 | 3.06 (0.76) |
| **CLAIR** | DPO | 0.55 | −1.68 (1.73) | 5.05 | 2.77 (1.40) |
| | KTO-pair | 2.15 | 0.79 (0.98) | 4.65 | 2.92 (0.86) |
| | SFT | 0.65 | −0.91 (1.01) | 2.70 | 0.92 (1.21) |
| | **APO-zero** | **7.65** | **2.93** (1.98) | **5.95** | **4.39** (0.89) |
| | APO-down | −1.05 | −5.22 (1.55) | −1.20 | −3.61 (1.05) |
| Stronger Preferred | DPO | −5.00 | −6.94 (1.03) | −3.10 | −4.40 (0.98) |
| | KTO-pair | −1.20 | −5.21 (1.27) | 2.25 | 0.50 (1.13) |
| | SFT | 2.45 | 0.49 (1.31) | 5.05 | 2.73 (1.21) |
| | APO-zero | −1.70 | −2.72 (1.40) | −4.85 | −12.02 (5.38) |
| | APO-down | −6.50 | −12.51 (4.97) | 1.65 | 0.16 (1.22) |

Table 1: Max and mean `MixEval-Hard` improvements for the `2024-06-01` and `2024-08-11` splits, aggregated over 18 epochs of aligning `Llama-3-8B-Instruct`. Best overall performance **bold**, best performance per dataset underlined, standard deviation in parentheses. CLAIR leads to the greatest overall performance improvement on `MixEval-Hard`. APO methods achieve the best performance across both Judged and CLAIR datasets.

# 5 Alignment Experiments

To study the effectiveness of CLAIR and APO, we align `Llama-3-8B-Instruct` across the four comparable preference datasets described in Section 3, created from 32K `UltraFeedback` prompts. We use GPT4-turbo to act as *Judge* or *Reviser* when creating these datasets. For every dataset, we align the model using the four different objectives described in Section 4. Additionally, we consider Supervised Fine-Tuning (SFT) on only the winning outputs as a baseline alignment objective.

We consider the downstream performance change of the model to be indicative of reward hacking. The more the model improved, the more the optimization realized the intended behavior. If the model actually worsens during training, we consider this a clear example of reward hacking.

## 5.1 Evaluation Methodology

Human judgments are ultimately the best indicator of how well a model is aligned with human preferences. Chatbot Arena [Chiang et al., 2024] uses thousands of pairwise human judgements to produce a ranking of model performance. However, collecting these judgments can be prohibitively expensive. To overcome this obstacle, we measure model performance through benchmarks which correlate highly with this Chatbot Arena ranking.

`MixEval-Hard` [Ni et al., 2024] is a benchmark with very high Chatbot Arena correlation (0.96 rank correlation). `MixEval-Hard` features hard queries with known answers across a wide range of domains and uses a GPT3.5-turbo [Brown et al., 2020, Ouyang et al., 2022] model to evaluate if predicted answers correspond with this ground-truth. This makes `MixEval-Hard` more grounded in human knowledge and significantly cheaper to run compared to other popular evaluation frameworks such as `AlpacaEval` [Li et al., 2023, Dubois et al., 2024].

Our evaluation of `Llama-3-8B-Instruct` before any additional alignment achieves a score of 41.45% on the `2024-06-01` version of `MixEval-Hard`. The gap between `Llama-3-8B-Instruct` and GPT4-turbo is 17%. On the `2024-08-11` split, `Llama-3-8B-Instruct` achieves 40.5%.

## 5.2 Training Specifications

`Llama-3-8B-Instruct` is trained for a total of 18 epochs on each preference dataset and alignment objective, with a checkpoint saved every single epoch. The $\beta$ hyperparameter, common to all alignment objectives except SFT, is set to 0.1. Prompt and responses are truncated to 512 tokens each. Each model is trained using an effective batch size of 16 across one node of 8 `NVIDIA H100` GPUs, using the `RMSProp` optimizer with a learning rate of $2 \times 10^{-7}$, linearly decaying to 0 over the 18 epochs. All training is implemented using the `TRL` library [von Werra et al., 2020].

## 5.3 Results

We report the maximal and mean `MixEval-Hard` improvement over all checkpoints from the same training run. This helps us understand both the best-case and average impact of alignment across the entire training procedure. We use both `2024-06-01` and `2024-08-11` versions of `MixEval-Hard`, which each feature a distinct set of queries. We use no system prompt. Our analysis is summarized in Table 1 for every dataset and objective; we now discuss these results in more detail.

### 5.3.1 Preference Data

To assess the quality of a particular dataset, we consider the performance of that dataset when paired with its best objective. Using the APO-zero objective, **the contrastive CLAIR dataset leads to the greatest improvement**, signalling the least amount of reward hacking. On the `2024-06-01` split of `MixEval-Hard`, CLAIR leads to the greatest maximal improvement of +7.65% and the greatest average improvement of +2.93% out of all our experiments. This improvement of +7.65% closes the relative gap with GPT4-turbo by 45% using only 32K pairs.

We noted in Section 1 that uncontrolled contrastiveness can degrade model performance. We see this dramatically in the results for the Stronger Preferred dataset, which can heavily degrade model performance. Like CLAIR, this dataset has all winning outputs produced by a stronger model. Unlike CLAIR, though, its examples provide no guarantee of relevant minimal contrasts. Thus, **the contrastiveness induced by the CLAIR revision process is a major driver of performance**.

Both on-policy judge and off-policy judge datasets lead to improved performance when paired with their best alignment objective, but **on-policy preferences lead to better performance compared to off-policy preferences**. This is intuitive; judgments about the target model's outputs are in general more relevant.

### 5.3.2 Alignment Objectives

On `MixEval-Hard`, **Anchored Preference Optimization (APO) consistently leads to the greatest performance increase for every preference dataset**, with the exception of the Stronger Preferred dataset, where all contrastive objectives underperform SFT. The relation between the preference dataset and the target model controls which variant of APO is best for any dataset, as predicted in Section 2. **APO-down results in the best performance when winning outputs are generally worse than the target model**, as is the case for the off-policy judge dataset. **APO-zero is the best objective when winning outputs are generally better than the target model**, as is the case for CLAIR and on-policy judge datasets. The difference between alignment objectives is less salient for the on-policy judge dataset as compared to CLAIR, since winning on-policy judge outputs are only slightly better than `Llama-3-8B-Instruct` on average. Winning CLAIR outputs may be vastly better than `Llama-3-8B-Instruct` since they are produced by a stronger model, making the different in alignment objectives more noticeable.

## 6 Related Work

We now characterize relevant alignment efforts and outline how they relate to Contrastive Learning from AI Revisions (CLAIR), Anchored Preference Optimization (APO), and reward hacking.

Reinforcement Learning from Human or AI Feedback (RLHF / RLAIF; Ouyang et al. 2022, Bai et al. 2022, Yuan et al. 2024) is a technique used to align models with human preferences. Fundamentally, these approaches first train a reward model using preference judgments and subsequently optimize a

Language Model for this reward using Reinforcement Learning [Schulman et al., 2017]. To side-step the need for an explicit reward model, Direct Preference Optimization (DPO; Rafailov et al. 2024b) aligns an LM directly using a contrastive training objective.

We articulated two core insights concerning alignment and reward hacking, specifically 1. the role of contrastive preference data, and 2. the need to anchor alignment depending on model and data. These insights translate to any alignment effort which uses comparative preferences. For example, a reward model trained on spurious preference signals may be a less accurate proxy for real rewards, directly causing reward hacking [Gao et al., 2023, Rafailov et al., 2024a].

For the remainder of this review, we first focus on contrastive alignment methods and their variants (of which Wang et al. 2024 provide a detailed overview). Finally, we discuss related preference datasets and how they were created.

**Controlling training dynamics:** The tendency of DPO to decrease the winning likelihood has been remarked and analyzed in several works [Feng et al., 2024, Pal et al., 2024]. Some works use an additional loss term to explicitly increasing the likelihood of winning outputs [Hong et al., 2024, Pentyala et al., 2024, Adolphs et al., 2023, Zhao et al., 2023, Xu et al., 2024]. While these methods can be seen as variants of Anchored Preference Optimization, they do not recognize the need to anchor the objective differently depending on dataset and model, and they do not offer methods that explicitly decrease the winning likelihood when required. Both Rafailov et al. [2024a] and Azar et al. [2024] generalize a set of alignment methods, but neither allow for any anchoring. We have found that this anchoring is crucial to mitigate reward hacking across several situations.

**Length-controlled optimization:** Preference pairs created through a judging paradigm can be biased towards preferring more verbose answers Saito et al. [2023]. This can be seen as a clear example of reward hacking. To prevent aligned models from inheriting this bias, Meng et al. [2024] and Park et al. [2024] explicitly control for the length of generations during training. These constraints on generation length can be seamlessly integrated into APO methods as well. In addition, CLAIR revisions could further help with these efforts to reduce the verbosity bias. For example, the *Reviser* could be designed to not increase length.

**Preference Datasets:** Chiang et al. [2024] release a dataset of human preference judgements across conversations between humans and several AI assistants. To alleviate the need for human judges, some efforts focus on scaling preference annotations with LLM-based judges [Cui et al., 2024, Zhu et al., 2023] or metric-based judges [Jiang et al., 2023]. Unlike our CLAIR method, these works do not create preferences through revisions. Bai et al. [2022] use a set of predetermined criteria (called a *constitution*) to prompt an LLM to revise answers and make them safer (see also Lambert et al. 2024). Dubey et al. [2024] used human revisions in the development of the `llama-3.1` model family. While both efforts create preferences through revisions, we particularly focus on revisions that create a minimal contrast and we studied the effect of this contrastiveness on alignment outcomes. We have found that preferences with less spurious correlations lead to significantly less reward hacking, in turn producing more performant models.

# 7   Conclusion

Language Model alignment is one of the primary tools used to develop safe and ethical models. Significant safety issues may occur when alignment training does not realize its intended behavior. In this work, we found that alignment performance is significantly impacted by 1. the contrastiveness of the preference pairs and 2. the relationship between target model and alignment data. We introduced Contrastive Learning from AI Revisions (CLAIR), a data-creation method which produces contrasting preference pairs with minimal spurious differences, and Anchored Preference Optimization (APO), a family of alignment objectives with tailored and more controlled training dynamics. Our experiments aligning `Llama-3-8B-Instruct` show that CLAIR preferences lead to the highest performance improvement out of four comparable preference datasets, and APO methods consistently outperform conventional alignment objectives. The improved performance of our methods indicates a more robust training procedure, better realizing the intended behavior of the alignment procedure.

## Acknowledgements

We thank Kawin Ethayarajh, Eugen Hotaj, and Nathan Lambert for their feedback. We thank Stas Bekman for his help and support. KD gratefully acknowledges funding from the FWO Fundamental Research PhD Fellowship (11632223N).

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

# A    Preference Dataset Creation

## A.1    Prompts

The prompts we use for the *Reviser* and *Judge* function of Equation 1 and 2 are given in Table **??**. Both prompts contain instructions to prefer more clear, more correct, and more engaging outputs. The *Reviser* prompt creates a preference pair by minimally revising and improving an output according to these preferences. Instead, the *Judge* prompt selects a more preferred output given two candidate answers.

## A.2    Preference Pair Filtering

We reject revisions or judgments if the LLM failed to follow formatting guidelines specified in the revising or judging prompt. Additionally, we reject revisions if they altered the length of the original output too much; we found this mainly happens when the LLM misunderstands the revision prompt. Starting from the same 32K instructions sampled from `UltraFeedback`, this procedure creates 29K CLAIR pairs, 29K Stronger Preferred pairs, 29K off-policy Judge pairs, and 32k on-policy Judge pairs. We adapted the code by Williams [2023] to efficiently query closed-source LLMs in parallel over API.

# B    `MixEval-Hard` Performance Breakdown

`MixEval-Hard` features queries from a wide range of established benchmarks. Under the hood, `MixEval-Hard` utilizes queries sampled from `MATH` [Hendrycks et al., 2021], `BBH` [Suzgun et al., 2023], `DROP` [Dua et al., 2019], `GSM8k` [Cobbe et al., 2021], `AGIEval` [Zhong et al., 2024], `TriviaQA` [Joshi et al., 2017], `MBPP` [Austin et al., 2021], `MMLU`, [Hendrycks et al., 2020], `HellaSwag` [Zellers et al., 2019], `BoolQ` [Clark et al., 2019], `GPQA` [Rein et al., 2023], `PIQA` [Bisk et al., 2020], `OpenBookQA` [Mihaylov et al., 2018], `ARC` [Clark et al., 2018], `CommonsenseQA` [Talmor et al., 2019], and `SIQA` [Sap et al., 2019]. Previously, we reported on the overall `MixEval-Hard` performance. Table 2 breaks down this overall performance in function of these different benchmarks. While `MixEval-Hard` often incorporates only a few queries from any given benchmark, the overall performance correlates highly with human judgements.

| MixEval-Hard split | # query | Llama-3-8B -Instruct | + CLAIR | + Judge (on-policy) | + Judge (off-policy) | + Stronger Preferred |
|---|---|---|---|---|---|---|
| Overall score | 988 | 41.45 | **49.10** | 46.10 | 44.15 | 43.90 |
| TriviaQA | 267 | 34.30 | **49.20** | 42.40 | 43.70 | 39.80 |
| MMLU | 231 | **43.70** | 39.00 | 42.00 | 36.80 | 34.60 |
| DROP | 167 | 50.20 | 58.70 | 64.30 | **64.90** | 58.90 |
| AGIEval | 71 | 31.00 | 38.00 | 38.00 | **39.40** | 38.00 |
| HellaSwag | 61 | 29.50 | **37.70** | 26.20 | 29.50 | 27.90 |
| CommonsenseQA | 50 | 60.00 | **72.00** | 60.00 | 48.00 | 58.00 |
| BoolQ | 37 | 40.50 | **45.90** | 32.40 | 21.60 | 27.00 |
| GSM8k | 22 | 60.00 | 80.00 | 69.50 | 63.20 | **84.10** |
| SIQA | 20 | 45.00 | **50.00** | 40.00 | 15.00 | 40.00 |
| MATH | 16 | 47.50 | 63.70 | 51.30 | 58.80 | **73.10** |
| BBH | 16 | 51.30 | **68.80** | 57.50 | 60.60 | 66.90 |
| OpenBookQA | 8 | 62.50 | 62.50 | 50.00 | 62.50 | **75.00** |
| GPQA | 8 | 12.50 | 25.00 | 25.00 | 25.00 | **37.50** |
| PIQA | 8 | 50.00 | 62.50 | 62.50 | 62.50 | **75.00** |
| ARC | 4 | **0.00** | **0.00** | **0.00** | **0.00** | **0.00** |
| MBPP | 2 | **0.00** | **0.00** | **0.00** | **0.00** | **0.00** |
| Objective used: | | / | APO-zero | APO-zero | APO-down | SFT |

Table 2: Breakdown of `MixEval-Hard` performance (version `2024-06-01`) in function of which dataset the queries originate from. Analysis given for `Llama-3-8B-Instruct` and our best models on the CLAIR, Judge (on-policy), Judge (off-policy), and Stronger Preferred datasets. While individual splits may not always indicate the best model (particularly when the amount of queries is low), the overall score correlates highly with human judgments about model performance (Chatbot Arena Elo; Chiang et al. 2024). `MixEval-Hard` uses a GPT3.5-turbo model to rate if a response to a query agrees with a known gold-truth response.

