# OpenReview forum: "Anchored Optimization and Contrastive Revisions: Addressing Reward Hacking in Alignment"
_NeurIPS.cc/2024/Workshop/SafeGenAi — SafeGenAi Poster_

### Official Review · Reviewer_CVpp · 2024-10-08
**APO effectively address reward hackling in alignment**

**Rating:** 9
**Confidence:** 5

**Review:**

The author meticulously analyzed the problems with current DPOs and provided an innovative and interesting solution——APO. This article has a high quality, originality, and excellent writing level, and it really solves some of the problems that currently exist in DPO.

---

### Official Review · Reviewer_Sn8B · 2024-10-09

**Rating:** 6
**Confidence:** 2

**Review:**

- This is a very interesting paper. However, my main concern with the paper is that the evaluations are only done using one model (llama-3-instruct) which is limiting.

- The idea of this paper is to mitigate reward-hacking by using contrastive examples as y_l and y_w.